# Increased artificial illumination delays urban autumnal foliar senescence

Yang Chen[1,2,7], Wendi Qu[3,7], Constantin M. Zohner [4], Josep Peñuelas[5,6] & Chaoyang Wu [1,2] ✉

Rapid urbanization has driven widespread increases in artificial light at night, intensifying energy use, light pollution, and sustainability challenges. However, its ecological impacts, particularly on vegetation phenological transitions, remain poorly understood. Using 62,994 site-year in situ records and satellite observations across 452 cities from 2001 to 2022, we show that elevated levels of artificial light at night are associated with delayed dates of foliar senescence in urban areas. This delaying effect is spatially heterogeneous and nonlinear, being most pronounced at low light intensities ($< 15$ nW cm$^{-2}$ sr$^{-1}$) and decreasing or saturating at higher levels. Regional variability in effects of artificial light at night is primarily shaped by urban socioeconomic factors and vegetation traits. Mechanistically, the delaying effect may result from enhanced carbon assimilation and altered climatic responses. We further improve the phenological modeling by incorporating the effects of artificial light at night and project overall later foliar senescence dates than currently predicted. Collectively, our findings highlight a previously underrecognized pathway by which urbanization alters vegetation phenology, with implications for forecasting ecosystem dynamics under continued urban growth and climate change.

Artificial light at night (ALAN) has increased substantially in intensity, spatial extent, and duration with the continued expansion of electrical infrastructure and increasing population density due to the acceleration of urbanization[1–3]. Satellite observations show that ALAN-affected land is growing at an annual rate of 2.2%[4]. Currently, approximately 83% of the global population lives under light-polluted skies, and nearly 23% of Earth's terrestrial surface is exposed to ALAN[2]. Beyond urban centers, ALAN intrudes into natural ecosystems via direct illumination and indirect scattering (e.g., skyglow), potentially impacting areas tens to hundreds of kilometers from city cores[5]. ALAN is known to disrupt animal behavior and phenology, such as altering bird migration[6] and insect reproduction[7], and even human circadian

rhythms[8], but its ecological impact on vegetation, particularly phenological responses, remain underexplored[9].

Vegetation autumn phenology, i.e., the date of foliar senescence (DFS), plays a vital role in regulating the carbon balance and nutrient cycling, yet it has received far less attention than spring phenology due to its complex and multifactorial controls[10–12]. Previous studies have primarily examined climatic drivers such as temperature[11,13], precipitation[14,15], and solar radiation[16], but the underlying mechanisms remain incompletely understood. This gap is particularly evident in urban environments[17], where anthropogenic disturbances, such as urban heat island effects[18], air pollution[19], and especially ALAN[9,20,21], may substantially modify vegetation dynamics. As an informational

[1]The Key Laboratory of Land Surface Pattern and Simulation, Institute of Geographical Sciences and Natural Resources Research, Chinese Academy of Sciences, Beijing, China. [2]University of the Chinese Academy of Sciences, Beijing, China. [3]School of Ecology and Nature Conservation, Beijing Forestry University, Beijing, China. [4]Department of Environmental Systems Science, Institute of Integrative Biology, ETH Zurich, Zurich, Switzerland. [5]CSIC, Global Ecology Unit CREAF-CSIC-UAB, Barcelona, Catalonia, Spain. [6]CREAF, Cerdanyola del Valles, Barcelona, Catalonia, Spain. [7]These authors contributed equally: Yang Chen, Wendi Qu. ✉e-mail: wucy@igsnrr.ac.cn

pollutant, ALAN interferes with photoreceptor perception of natural light cues, especially when emitted from sources rich in red and far-red wavelengths (e.g., high-pressure sodium and incandescent lamps), leading to abnormal signal transduction and physiological dysfunction[22]. High-intensity ALAN may cause photoinhibition, and under drought or nutrient stress, even moderate light levels can result in energy imbalance and light-induced damage, consistent with the "excess light" theory[23]. However, quantifying and disentangling the influence of ALAN on DFS remains challenging. Empirical studies based on manipulative experiments and satellite observations have reported inconsistent results due to variations in phenological indicators, spatial scales, and quantitative approaches[20,21,24–26], thereby hindering the identification of generalizable patterns and mechanisms. For example, a recent satellite-based study reported that elevated ALAN delayed DFS to a greater extent than urban warming[20], whereas another found that ALAN altered vegetation climatic responses and advanced DFS[21]. A leading hypothesis attributes ALAN effects on DFS to changes in plant carbon sink activity and capacity[27,28], whereby ALAN modulates photosynthesis and biomass allocation, ultimately influencing senescence timing. Given the accelerating pace of urbanization and climate change, cross-scale and systematic assessments are urgently needed to reduce uncertainties in phenological forecasts and to inform the development of ALAN-integrated DFS models[24].

In this study, we analyze 452 major urban areas in mid-to-high latitudes from 2001 to 2022. We integrate multi-scale observations, including 62,994 in situ DFS records from Europe and China, satellite-derived ALAN and DFS data, and climate variables (Supplementary Table 1), to quantify the impacts of ALAN on urban DFS. We also examine the spatial drivers of ALAN's effects, explore potential mechanistic pathways linking ALAN to DFS changes, and improve phenological models by incorporating ALAN effects to project future DFS trends under various scenarios.

## Results

### Response of DFS to increased ALAN intensity

Distinct spatial patterns were observed in both DFS and ALAN intensity across cities. DFS occurred earlier at higher latitudes, while ALAN intensity was greater in developed regions such as North America, Europe, South Korea, and Japan (Supplementary Fig. 1a, c). Temporally, DFS was delayed in 343 cities (75.9%) from 2001 to 2022, with an average trend of 0.77 days per year (Supplementary Fig. 1b). Concurrently, ALAN increased in 430 of the 452 cities (95.1%), with a mean rate of 0.53 nW cm$^{-2}$ sr$^{-1}$ y$^{-1}$ (Supplementary Fig. 1d).

We identified a predominantly positive effect of ALAN on DFS, with 304 cities (67.3%) showing positive ALAN sensitivity (SV$_{ALAN}$), suggesting that increased ALAN generally delays DFS. This delaying effect was particularly pronounced in Chinese cities, whereas cities in Europe and North America showed a more balanced distribution of positive and negative SV$_{ALAN}$ values (Fig. 1a). A nonlinear, negative relationship between SV$_{ALAN}$ and ALAN intensity was identified, best described by a Gaussian function ($y = 0.24e^{\frac{-(x-3.39)^2}{35.36}}$, $R = -0.60$, $P = 0.00$) (Fig. 1b), indicating that the delaying effect weakens when ALAN intensity exceeds 15 nW cm$^{-2}$ sr$^{-1}$. Climatically, ALAN tended to delay DFS more strongly in warmer and drier cities (Fig. 1c). To validate the satellite-based findings, we calculated SV$_{ALAN}$ using ground-based DFS observations from Europe and China (Supplementary Fig. 2). The proportions of sites or pixels with positive and negative partial correlations were broadly consistent for in situ and satellite estimates, particularly confirming dominant delaying effects in China (Fig. 1d). Similar ALAN-DFS patterns were observed at national, city buffer, and site scales (Supplementary Figs. 3–5). To ensure the robustness of our results, we compared multiple independent ALAN products (Supplementary Figs. 6 and 7) across different time periods, which revealed highly consistent spatial and statistical patterns. Finally, using the Peter–Clark Momentary Conditional Independence Plus (PCMCI+)

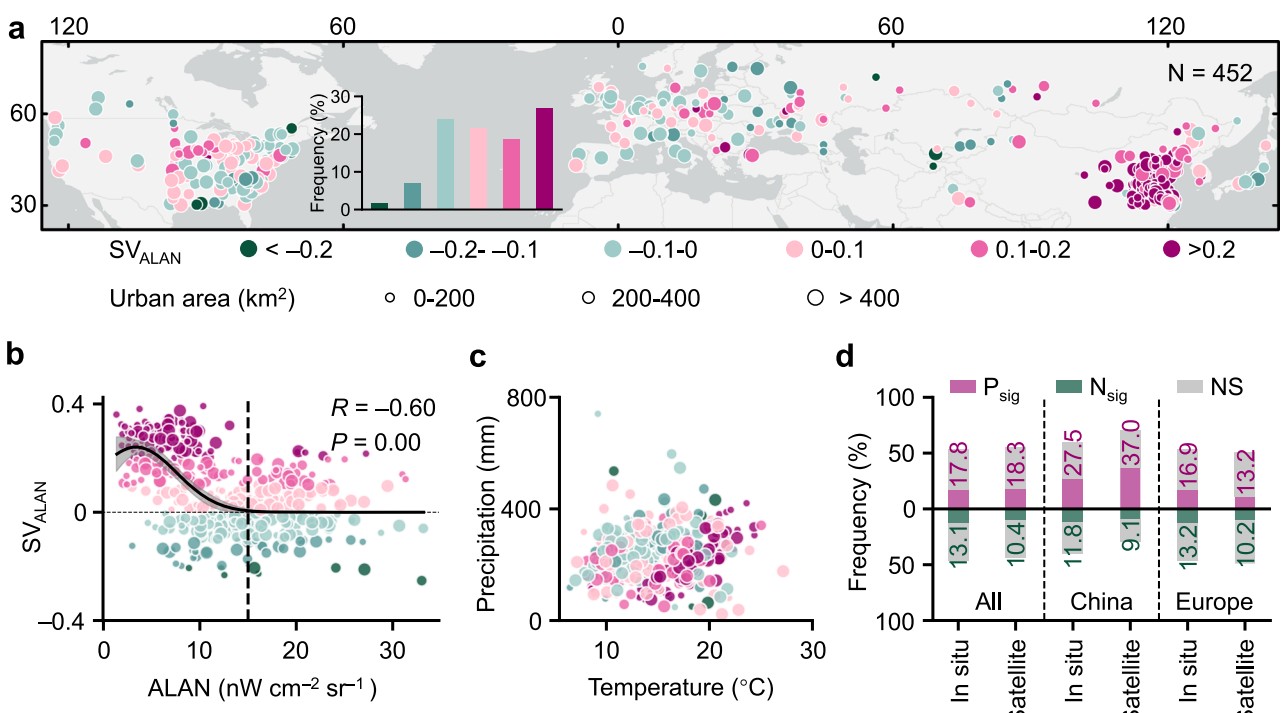

**Fig. 1 | Spatial patterns and climatic controls of the sensitivity (SV$_{ALAN}$) of the date of foliar senescence (DFS) to the intensity of artificial light at night (ALAN). a** Spatial patterns of SV$_{ALAN}$. N represents the number of cities. **b** Trend of SV$_{ALAN}$ with increasing ALAN intensity. The black line shows the Gaussian fit to the data. The gray shaded area indicates the 95% confidence intervals of the fit.

($P < 0.001$, two-tailed $t$-test) **c**, Variation in SV$_{ALAN}$ with mean temperature and precipitation. **d**, Distribution of partial-correlation coefficients between ALAN and DFS for in situ and satellite observations. P$_{sig}$, N$_{sig}$, and NS represent significantly positive, negative, and nonsignificant correlations, respectively ($P < 0.1$, two-tailed $t$-test). Source data are provided as a Source Data file.

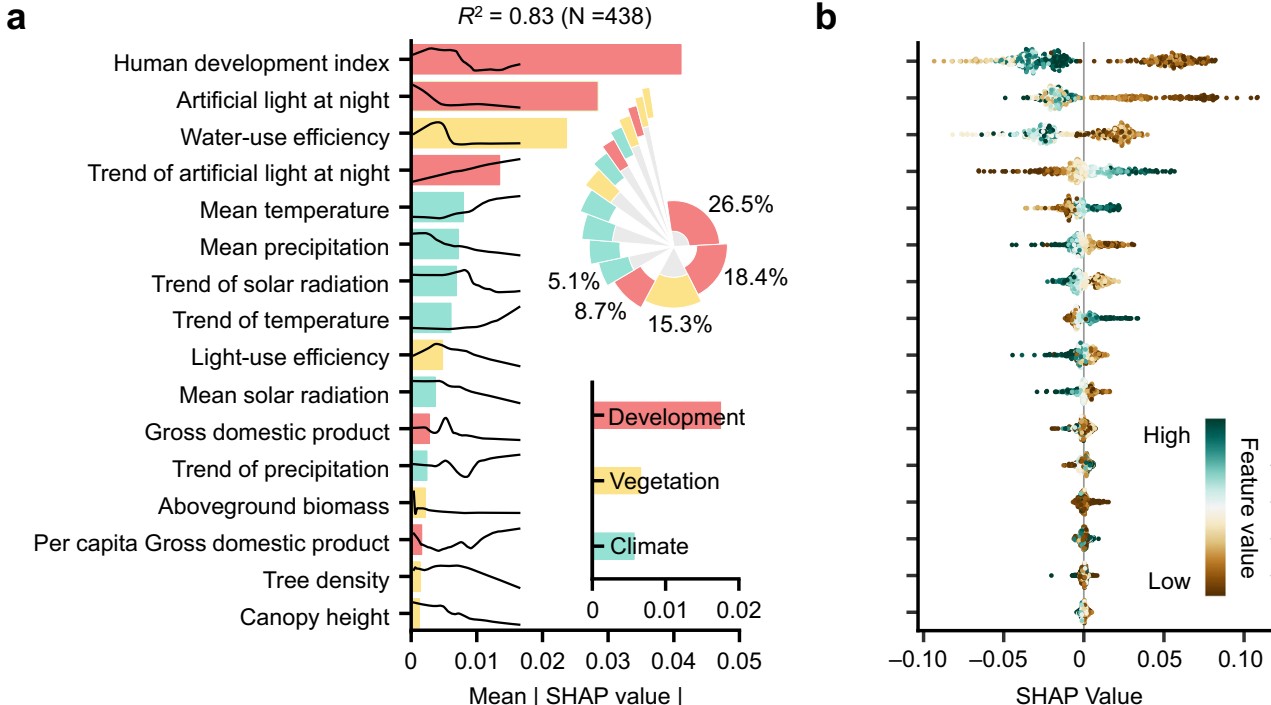

**Fig. 2 | Spatial attribution analysis of artificial light at night (ALAN) effects on the date of foliar senescence (DFS). a** The relative importance of factors controlling the spatial variability of ALAN effects on DFS (i.e., $SV_{ALAN}$), determined by a random-forest model ($R^2 = 0.83$, $N = 438$) using mean absolute SHAP values. The inner subplot indicates the average importance of factors associated with development, vegetation, and climate. **b** The right beeswarm plot shows the distribution of SHAP values for each factor. Source data are provided as a Source Data file.

method (Methods), we identified directional causality from ALAN to DFS in 48.3% of the time series, which was more frequently than for climatic drivers (temperature, precipitation, shortwave radiation) (Supplementary Fig. 8).

**Spatial attribution analysis of ALAN effects**

We applied an eXtreme Gradient Boosting (XGBoost) model combined with Shapley Additive Explanations (SHAP) to assess the relative importance of factors driving the spatial variability in ALAN effects on DFS, i.e., $SV_{ALAN}$ (Methods). All driving factors explained 83% of the spatial variability ($R^2 = 0.83$, $N = 438$). The most influential predictors included the human development Index (HDI), ALAN intensity and its temporal trend, water-use efficiency (WUE), and long-term mean temperature and precipitation. We categorized the variables into three groups, i.e., socioeconomic development, climatic conditions, and vegetation properties, and found that socioeconomic factors contributed the most explanatory power (Fig. 2a). SHAP values indicated that regions with lower HDI or lower ALAN levels were more likely to exhibit positive ALAN effects on DFS (Fig. 2b). Additionally, areas with higher mean temperatures and lower precipitation showed stronger positive DFS-ALAN relationships, consistent with earlier results (Figs. 1c and 2a).

**Potential mediating processes underlying the ALAN-DFS relationship**

Through an extensive review and synthesis of relevant literature, we proposed two hypotheses to investigate the mechanisms by which increases in ALAN intensity could delay DFS (Supplementary Table 2): (H1) increased ALAN enhances vegetation photosynthetic carbon fixation, thereby postponing DFS; (H2) ALAN alters the response of DFS to climatic drivers, further affecting its temporal variation.

To test H1, we conducted partial correlation analyses using two proxies for photosynthetic capacity: the maximum rate of

carboxylation ($V_{cmax}$) and solar-induced chlorophyll fluorescence (SIF). Among the 304 cities exhibiting delaying effects of ALAN, 146 (48%) had a significant positive ALAN-$V_{cmax}$ correlation, while only five cities (1.6%) showed a negative one ($P < 0.05$). Of these 146 cities, 108 (74%) also showed a significant positive $V_{cmax}$-DFS correlation (Fig. 3a). Analyses based on SIF similarly supported H1, though with weaker spatial consistency (Fig. 3b, Supplementary Fig. 9). We further used the PCMCI+ method to assess causal links between ALAN, photosynthesis proxies, and DFS, which confirmed notable causal pathways (Supplementary Fig. 10). To test H2, we applied a moving-average approach (Methods) to examine how ALAN modulates the sensitivity of DFS to temperature, precipitation, and shortwave radiation. Higher ALAN exposure generally enhanced temperature sensitivity (32.2% positive vs. 14.1% negative; $P < 0.05$; Fig. 3c). Given the widespread warming trends and the delaying effects of warming on DFS across cities (Supplementary Figs. 11a and 12a), ALAN-amplified temperature sensitivity may further delay DFS. In contrast, ALAN reduced precipitation sensitivity (14.1% positive vs. 26.3% negative; $P < 0.05$; Fig. 3c). Considering the overall decreasing trend in precipitation and its delaying effects on DFS (Supplementary Figs. 11b and 12b), weakened precipitation sensitivity under elevated ALAN may also contribute to delayed DFS. ALAN showed no consistent effect on radiation sensitivity, with comparable proportions of positive (19.4%) and negative (22.7%) correlations (Fig. 3c). When cities were grouped by ALAN intensity, both hypotheses received stronger support in low-ALAN regions ($< 15\,nW\,cm^{-2}\,sr^{-1}$), underscoring the nonlinear nature of ALAN's influence (Supplementary Fig. 13). Collectively, these results support both hypotheses and provide mechanistic insights into how increased ALAN delays DFS.

**DFS modeling and prediction**

We improved DFS models by incorporating ALAN effects using four widely used models: the semi-empirical model for cooling degree days

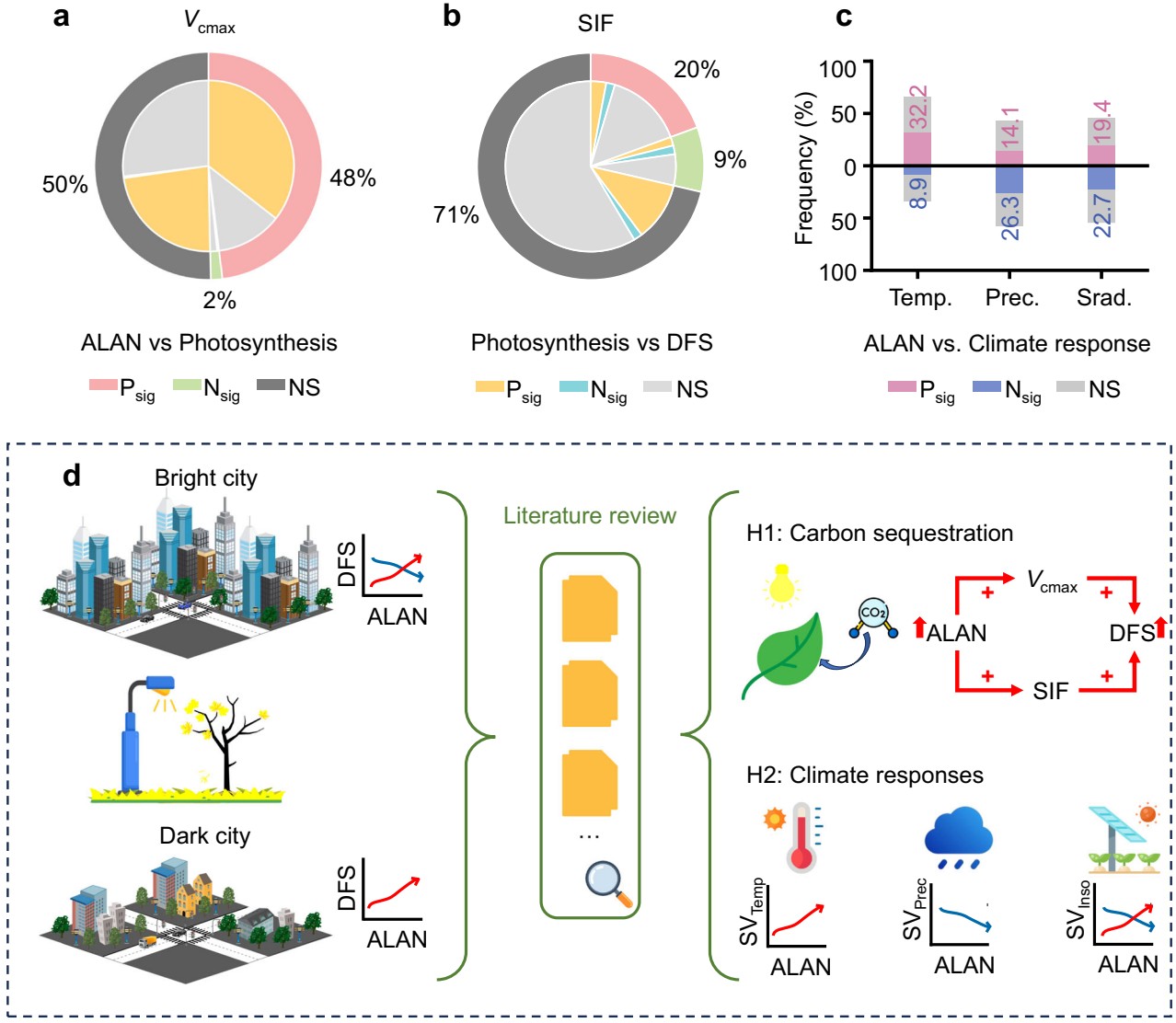

**Fig. 3 | Mediating pathways linking artificial light at night (ALAN) to the date of autumnal foliar senescence (DFS). a, b,** Proportions of cities with significantly positive ($P_{sig}$), negative ($N_{sig}$), and nonsignificant (NS) partial correlations between ALAN and photosynthesis (outer rings) and between photosynthesis and DFS (inner rings) ($P < 0.05$, two-tailed $t$-test). Photosynthetic indicators include the maximum rate of carboxylation ($V_{cmax}$) (**a**), solar-induced chlorophyll fluorescence (SIF) (**b**). **c** The modulating effect of ALAN on climatically driven DFS responses. **d**, Conceptual diagram illustrating potential mechanisms by which ALAN influences DFS variations. Source data are provided as a Source Data file.

(CDD), the Delpierre Model (DM), the spring-influenced autumn DFS model (SIAM), and the DM modified by spring-summer temperature (DMT) (Methods, Supplementary Fig. 14). Incorporating ALAN substantially enhanced model performance at both site and city levels, as evaluated by correlation coefficients ($R$), the proportion of sites or pixels with significant correlations, root mean square error (RMSE), and Akaike information criterion (AIC) (Fig. 4, Supplementary Figs. 15–17). For site-level data, SIAM with ALAN ($SIAM_{ALAN}$) achieved the highest $R$ (0.32), a 28% improvement over the original SIAM ($R = 0.25$). For satellite-derived data, $DMT_{ALAN}$ performed best ($R = 0.36$), nearly doubling the performance of the original DMT ($R = 0.19$). The increased frequency of significant correlations and reduced RMSE confirmed the improved DFS estimates after incorporating ALAN. Additionally, lower AIC values in ALAN-enhanced models indicated better model fit without overfitting, despite the inclusion of new parameters.

We further used DMT and $DMT_{ALAN}$ to project future DFS under two climatic scenarios: Shared Socioeconomic Pathways (SSP) 245 and 585 (Methods). Compared to the original DMT, $DMT_{ALAN}$ projected consistently later DFS dates. During 2081–2100, $DMT_{ALAN}$ estimated delays of $2.46 \pm 9.73$ days under SSP245 and $2.3 \pm 12.05$ days under SSP585 relative to the DMT ensemble mean. Moreover, $DMT_{ALAN}$ projections indicated significant delaying trends in DFS over 2001–2100, with slopes of 0.147 and 0.224 days per year under SSP245 and SSP585, respectively ($P < 0.001$; Fig. 4e).

## Discussion

Investigating the influence of ALAN on DFS provides critical insights into how urbanization alters plant phenology by modifying the light environment, thereby advancing our understanding of carbon cycling and climate adaptation mechanisms in urban ecosystems. Using both in situ and satellite observations, we demonstrate that increased ALAN delays DFS in urban vegetation across the Northern Hemisphere, with this effect observed in approximately two-thirds of the analyzed cities. This finding aligns with previous studies conducted across diverse spatial scales and ecological contexts[20,25]. The ALAN-DFS relationship followed a nonlinear pattern, weakening and saturating at higher light intensities. In regions with intense ALAN, the delaying effect

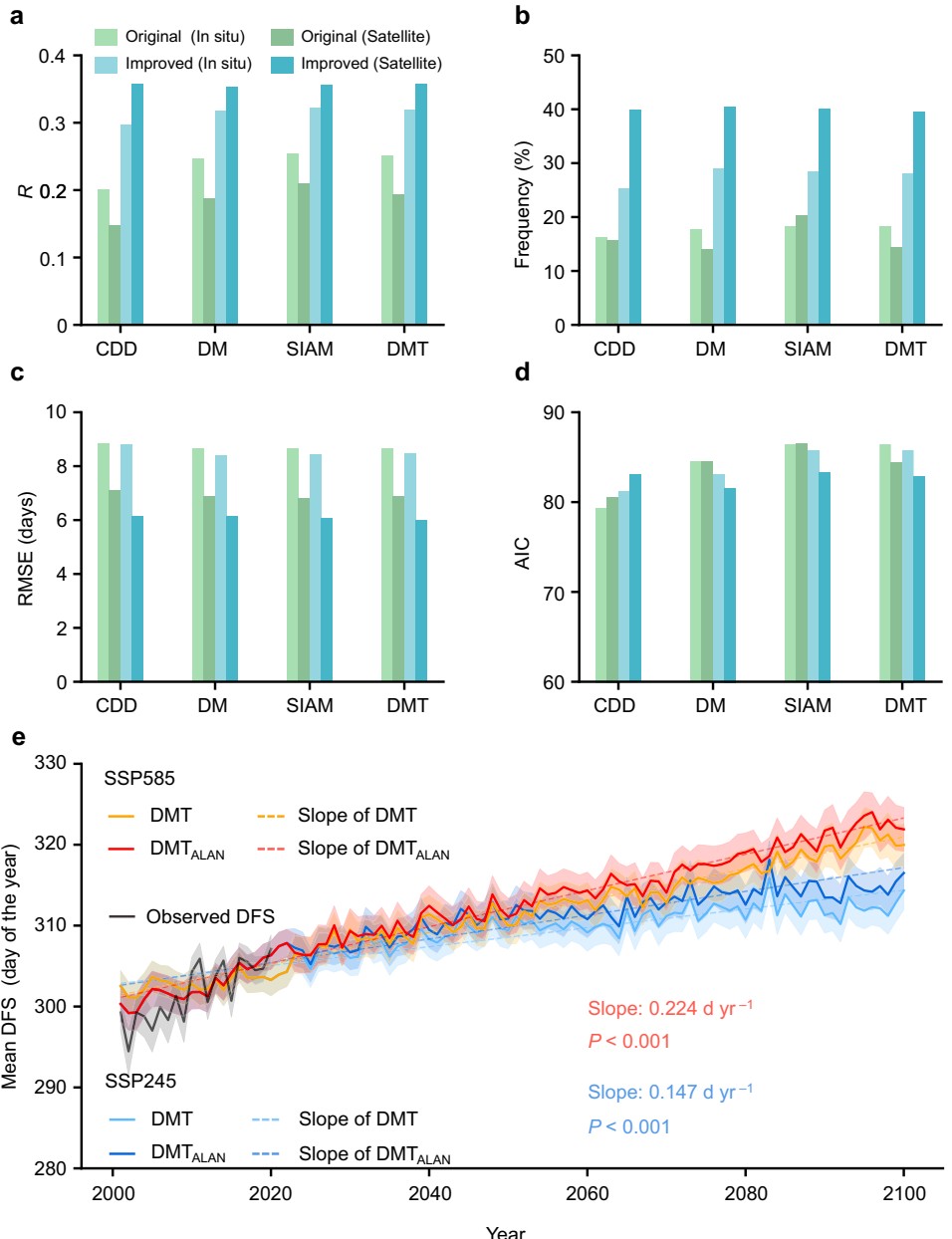

**Fig. 4 | Model comparisons with and without the consideration of the effects of artificial light at night (ALAN). a–d** The criteria for evaluating the models include the average correlation coefficients ($R$, **a**), the frequency of grids with significant correlation (**b**), the average root mean square error (RMSE, **c**) and the Akaike information criterion (AIC, **d**). Significance was set at $P < 0.05$. A two-sided $t$-test was used to assess the significance of the partial correlation analysis. The legend in (**a**) applies to **a**–**d**. The number in bracket represents the number of cities and sites. **e** Temporal trends of predicted DFS (2001–2100). Observed DFS is shown as a black line. Data are presented as mean values (bold lines) ±95% confidence intervals (error bands). Source data are provided as a Source Data file.

diminished, potentially because light-induced stress inhibits plant growth and counteracts the extended photoperiod that would otherwise postpone senescence[23]. We further found that ALAN-induced delays were more pronounced in warm, arid regions and attenuated or absent in colder areas, consistent with evidence that ALAN's phenological effects are temperature dependent[20]. This spatial heterogeneity suggests that ongoing climate warming could amplify the impact of ALAN on DFS. Moreover, cities with lower development and ALAN intensity tended to exhibit stronger positive DFS responses to ALAN. Technological transitions in street lighting, from high-pressure sodium lamps to energy-efficient white LEDs, have reduced energy use but simultaneously increased ALAN intensity and altered its spectral composition[29]. Developed regions are more likely to adopt blue-

enriched LED technologies, while less developed areas often retain older lighting systems that emit more red and far-red wavelengths. Differences in spectral structure and radiative characteristics among lighting types may therefore lead to divergent impacts on plant phenology[25].

Exploring the temporal linkage between ALAN and DFS presents additional challenges. Through an extensive review and synthesis of relevant literature, we proposed and examined two potential mechanisms through which ALAN delays DFS, related to carbon assimilation and climate responses. First, ALAN was positively associated with two photosynthetic proxies, i.e., $V_{cmax}$ and SIF, and higher $V_{cmax}$ and SIF positively correlated with delayed DFS. This finding supports the "extended photosynthesis" hypothesis[28,30,31]: despite low

levels of photosynthetically active radiation, ALAN may stimulate Photosystem II activity and Rubisco function, increase sugar accumulation, and delay senescence signals like ethylene synthesis[28]. The observed delays in DFS driven by enhanced growing-season productivity are consistent with evidence from eddy-covariance flux measurements[32], free-air $CO_2$ enrichment (FACE) experiments[33], and large-scale simulations[34]. Moreover, ALAN altered the sensitivity of DFS to temperature and precipitation. Elevated ALAN amplified the delaying effect of temperature, likely through circadian disruption mediated by photoreceptors like phyA and phyB[35]. Temperature-responsive genes can be upregulated under prolonged low light, enhancing plant responsiveness to warmth and delaying senescence[36]. In contrast, ALAN tended to reduce DFS sensitivity to precipitation, possibly because ALAN-induced stomatal opening enhances water loss, leading to chronic mild water stress that dampens vegetation responsiveness to short-term precipitation variability[15,37]. While these findings provide mechanistic insights into ALAN-phenology linkages, we acknowledge the limitations of analyses based solely on remote-sensing ALAN data. Future research should include controlled ALAN exposure experiments to directly measure hormonal responses, such as changes in auxin, ethylene, and abscisic acid, to better elucidate the physiological pathways underlying these effects[20].

Integrating ALAN effects into DFS models consistently and substantially improved predictive accuracy across scales. underscoring the importance of including ALAN when evaluating urbanization's effects on biogeochemical processes. As an emblem of urbanization, ALAN has ecological impacts that extend far beyond illumination. Our study systematically identified the nonlinear mechanisms and spatial heterogeneity of ALAN's influence on DFS, offering key insights into vegetation responses under global change. However, current models mainly capture direct photoperiodic effects of ALAN and often overlook indirect physiological pathways involving photosynthesis, hormonal regulation, and circadian rhythms. This limits our ability to fully understand its ecological consequences. Future research should integrate multi-scale observations, controlled experiments, and process-based modeling to better characterize ALAN's ecological effects and support the co-optimization of urban development and ecosystem conservation[38].

## Methods
### Site-level ground DFS data
This study integrated in-situ records of DFS from two authoritative networks of phenological observations: the China Phenological Observation Network (CPON)[39] and the Pan European Phenology Project (PEP725)[40]. We identified and removed potential outliers using median absolute deviation (MAD) method, which is more robust to skewed distributions and extreme values than standard deviation-based approaches. Specifically, MAD for a site- and species-specific DFS data set ($DFS_1$, $DFS_2$, …, $DFS_i$) was calculated as:

$$MAD = median\left(|DFS_i - median(DFS)|\right) \quad (1)$$

Any DFS value that exceeded 2.5 times the MAD was considered an outlier and excluded from further analysis[34]. We focused on DFS records from 2001 onward. We excluded site-species DFS time series with fewer than 10 years of data to ensure a robust analysis based on sufficient observations. We thereby obtained 5113 DFS records representing 228 species at 10 sites across China from CPON (2001–2018) and 57,881 DFS records representing six species at 1209 sites across Europe from PEP725 (2001–2015). See Supplementary Fig. 2 for details of the distribution of the observation sites.

### Satellite-derived DFS data
This study also used the Land Cover Dynamics phenological product (MCD12Q2 Version 6.1) of the Moderate Resolution Imaging Spectroradiometer (MODIS)[41]. These phenological metrics are derived from time series of the two-band enhanced vegetation index (EVI2). Spring phenology is defined as the first date when EVI2 exceeds 15% of its amplitude, and autumnal phenology is defined as the last date when EVI2 falls below 15% of its amplitude. These data offer crucial information for analyzing and monitoring vegetation phenological changes. We extracted the Greenup and Dormancy layers from the MCD12Q2 data set for 2001–2022, which represent the leaf-unfolding date (LUD) and DFS, respectively.

Satellite-based DFS estimates rely on seasonal changes in vegetation greenness (i.e. EVI2), which might not perfectly match ground-based measurements because of differences in spatial scale and pixel-mixing impacts. Specifically, MODIS EVI2 mixed pixels often incorporate multiple land cover types (e.g. forest, cropland, bare soil), while ground observations reflect localized homogeneous vegetation conditions. To address these potential biases, we analyzed the satellite and in situ data sets independently rather than through direct integration or comparison[8].

### ALAN data
We used the annual ALAN data derived from the "NPP-VIIRS-like" nighttime light product[42], which was generated by harmonizing observations from two satellite sensors, the Defense Meteorological Satellite Program Operational Linescan System (DMSP-OLS) and Suomi National Polar-orbiting Partnership Visible Infrared Imaging Radiometer Suite (NPP-VIIRS), using an improved autoencoder-based neural-network model. The NPP-VIIRS-like dataset exhibits strong ability to capture both population density distributions and changes in nighttime illumination across multiple spatial and temporal scales, closely resembling the composited NPP-VIIRS NTL data[42]. In addition, we also employed the original DMSP-OLS[43] and NPP-VIIRS nighttime light datasets[44], as well as another long-term harmonized nighttime light dataset (H-NTL-v2)[45]. Unlike the NPP-VIIRS-like dataset, the H-NTL-v2 dataset was generated by calibrating nighttime light data through the conversion of NPP-VIIRS observations into DMSP-OLS-like data. These complementary datasets were used to assess the consistency of DFS responses to ALAN across different data sources.

To better characterize the seasonal variations of ALAN intensity and its impact on DFS, we determined the monthly ALAN considering the potential duration of ALAN exposure[46]. We first calculated the solar declination-hour angle based on the latitude and day of the year (DOY) for each pixel to calculate the theoretical daily night length under cloud-free conditions. We then calculated the monthly average theoretical night length for each pixel as the duration exposed to ALAN. Finally, the monthly ALAN was calculated as:

$$ALAN_{month} = \frac{NH_{month} \times 12}{\sum_{month=1}^{12} NH_{month}} \times ALAN_{year} \quad (2)$$

where $ALAN_{month}$ represents the monthly ALAN value, $NH_{month}$ represents the average nightly hours per month and $ALAN_{year}$ represents the yearly ALAN value.

### Climatic and other ancillary data
We obtained monthly data for temperature, precipitation, and short-wave radiation for from the TerraClimate data set[47]. For DFS modeling, we used six-hourly temperature data from the CRU JRA v2.4 data set[48] and converted it into daily temperature data. To predict DFS under future scenarios, we also obtained daily temperature from the CMIP6 NorESM2-MM data set under both SSP245 and 585 scenarios[49]. We used the Global Artificial Impervious Area product to delineate urban boundaries[50]. Cities with built-up areas >100 km² were selected[20]. Ultimately, 452 major cities in the Northern Hemisphere were identified. Buffer zones covering 100, 200, and 400% of the urban areas were

generated to analyze the distance-decay effect of ALAN influence (Supplementary Fig. 4).

We conducted a spatial attribution analysis of the ALAN-DFS relationship by incorporating multiple factors of economic development and vegetation, including the Human Development Index, gross domestic product, per capita GDP, water-use efficiency, light-use efficiency, aboveground biomass, tree density, and canopy height. The data for HDI, GDP, and per capita GDP were obtained from the gridded global data sets for GDP and HDI[51]. For future scenarios, changes in per capita GDP relative to current statistical values were set based on country-specific scenarios from the llASA-SSP database[52]. We also used vegetation data, including aboveground biomass[53], tree density[54], and canopy height[55]. The WUE and LUE data represented the average monthly values from 2001 to 2022. Monthly WUE and LUE were calculated as:

$$WUE = \frac{GPP}{ET} \quad (3)$$

$$LUE = \frac{GPP}{SW \times 0.45 \times FPAR} \quad (4)$$

where GPP is gross primary productivit[56], ET is evapotranspiration[57], and FPAR is the fraction of photosynthetically active radiation[58], they are all derived from the MODIS products. SW is shortwave radiation derived from the TerraClimate monthly data set.

For the temporal analysis, we used the time-series data for $V_{cmax}$, and SIF as mediators to link ALAN and DFS. $V_{cmax}$ data from were obtained from the data set of maximum rate of carboxylation ($V_{cmax25}$)[59] and the SIF data were sourced from the global 'OCO-2' SIF data set[60]. See Supplementary Table 1 for detailed information on data descriptions, spatial and temporal resolutions, temporal coverage, and data sources for all data sets used in this study.

## Analysis

We applied the Theil-Sen trend analysis to quantify temporal changes in DFS, ALAN, and climate events. As a robust and nonparametric method, the Theil-Sen estimator is resistant to outliers and measurement errors, making it well-suited for long-term time-series analysis. To assess the statistical significance of these trends, we employed the Mann-Kendall test at a 0.05 significance level. This test is widely used because it does not assume normality or linearity and is unaffected by missing data or outliers.

The relationship between DFS and specific influencing factors in this study involved both simple correlation and partial-correlation analyses. We defined the influence of a factor on DFS as the maximum correlation coefficient between the phenological date and the variable values of different preseason lengths during the study period. We determined this influence by first calculating the optimal preseason length for each variable relative to DFS. Specifically, we used an exhaustive approach: starting from the month of the multi-year average phenological date, we extended backward in one-month steps up to six months, calculating the mean of each variable for these preseason periods[61]. We then performed a partial-correlation analysis between these means and the phenological time series to identify the preseason length that yielded the highest partial-correlation coefficient. Influencing factors often interact, so a partial-correlation analysis is typically used to isolate the effect of a single factor on DFS by controlling for the influence of other variables. Before performing correlation analyses, linear detrending is usually applied to each variable individually to remove long-term temporal trends, allowing the study to focus on the impact of the interannual variability of influencing factors on the interannual changes in DFS.

We employed a moving average approach to analyze the correlation between ALAN and climate sensitivity. Specifically, during the period from 2001 to 2022, we used an 11-year moving window with a 1-year step to calculate the average values of ALAN and climate sensitivity within each window, resulting in 12 data sets. Based on these moving averages, we then calculated the correlation coefficient to evaluate the relationship between ALAN and climate sensitivity. To avoid potential multicollinearity among the influencing factors, we employed ridge regression. Ridge regression is a linear regression model with L2 regularization, which introduces a penalty term in the loss function to suppress large regression coefficients, thereby effectively alleviating the problem of collinearity among independent variables. This method enhances the stability and generalization ability of the model. In this study, the response variable is DFS, and the predictor variables include preseason ALAN and climatic factors. We used the normalized anomalies of climatic factors, ALAN, and DFS as the inputs for the regression model. The resulting regression coefficients are interpreted as sensitivities of drivers.

To assess the potential causal influence of ALAN on the DFS and to further investigate whether ALAN, the photosynthesis indicators (SIF, $V_{cmax}$), and DFS exhibit not only statistical associations but also potential causal relationships, we employed the PCMCI+ algorithm to determine the direction of causality. Compared to traditional methods such as Granger causality that focus on bivariate prediction-based relationships, PCMCI+ can handle high-dimensional multivariate time series and account for complex interdependencies among multiple variables through conditional independence testing. PCMCI consists of two key steps: the PC algorithm and the Momentary Conditional Independence (MCI) test[62,63], which aimed at addressing the common issue of autocorrelation in time series data. In this study, we used the extended PCMCI+ framework, which is capable of identifying both lagged and contemporaneous causal relationships. Prior to applying PCMCI+, we first conducted partial correlation analysis to assess the statistical associations among ALAN, the photosynthesis indicators, and DFS. Subsequently, PCMCI+ was applied to infer whether these associations reflected potential causal structures while accounting for interdependencies among variables. It should be noted that PCMCI+ identifies statistically inferred causal relationships based on conditional independence, which provides evidence for potential causal mechanisms but does not establish definitive causation. All analyses were performed using the Tigramite Python package.

## Spatial attribution analysis of the ALAN-DFS relationship

We used interpretable machine learning with Shapley Additive Explanations (SHAP) to identify the key drivers of the spatial distribution of the effects of ALAN, quantified as the sensitivity of DFS to ALAN[64]. Various factors were categorized into three groups: (1) the economic-development factors HDI, GDP, per capita GDP, and ALAN; (2) the climatic factors temperature, precipitation, and shortwave radiation; and (3) the vegetation factors WUE, LUE, aboveground biomass, tree density, and canopy height. Both averages and trends were calculated for multi-year ALAN and the climatic variables. Detailed descriptions of all variables are provided in Supplementary Table 1.

We used these factors as predictor variables to develop an eXtreme Gradient Boosting (XGBoost) model. XGBoost is an optimized gradient boosting decision tree algorithm that is widely used in regression, classification, and ranking tasks due to its high computational efficiency and excellent predictive performance[65]. XGBoost effectively handles the complex nonlinear relationships and interactions commonly found in spatial environmental data and is highly robust to multicollinearity and missing values. In addition, its built-in regularization mechanism helps prevent overfitting, which is particularly important for modeling highly heterogeneous spatial data. Combined with efficient parallel computing capabilities, XGBoost can quickly and reliably process large-scale spatial datasets, enhancing the accuracy and interpretability of attribution analysis. To better interpret the predictions made by XGBoost, we employed the SHAP

method to quantify the marginal contribution of each predictor variable to the target variable. SHAP, a unified interpretation framework based on the game-theoretic Shapley value, fairly allocates the contribution of each feature to the model's predictions. It not only explains individual predictions but also reveals the overall importance of variables, greatly improving the transparency and trustworthiness of complex "black-box" models. These methods were implemented in Python using the "scikit-learn", "xgboost", and "shap" packages[65].

### Models for predicting DFS

We improved four widely used DFS models, i.e., CDD[66], DM[67], SIAM[68], and DMT[69,70]. Their respective versions enhanced with ALAN are termed $CDD_{ALAN}$, $DM_{ALAN}$, $SIAM_{ALAN}$, and $DMT_{ALAN}$.

The CDD model estimated DFS based on cumulative cold temperatures, as such conditions are likely key environmental signals triggering leaf coloration or senescence:

$$CDD_t = \sum_{t=t_0}^{t_y} max\left(T_{th} - T_{(t)}, 0\right) \quad (5)$$

$$DFS = t_r, \text{ if, } CDD_t \geq CDD_{th} \quad (6)$$

where $CDD_t$ denotes the cumulative cooling degree days calculated from the start date $t_0$ to a given day $t$, while $T(t)$ represents the average daily temperature on day $t$. The calculation begins on day $t_0$, defined as the first day when the temperature drops below the critical threshold ($T_{th}$), and continues until day $t_y$, when the total $CDD_t$ reaches a predefined value ($CDD_{th}$). The day $t_y$ is then recorded as the predicted DFS, represented by $DOY_{ty}$. In our study, the $T_{th}$ was set within a range of 0 to 50 °C.

Building upon the CDD model, the DM approach integrates both temperature and photoperiod as key factors influencing DFS:

$$S_{sen}(t) = S_{sen}(t-1) + R_{sen}(t) \quad (7)$$

$$R_{sen}(t) = \begin{cases} \left[T_b - T_{(t)}\right]^x \times f\left[P_{(t)}\right]^y, \text{ if } t \geq DOY_s \\ 0, \text{ if } t < DOY_s \end{cases} \quad (8)$$

$$f\left[P_{(t)}\right] = \frac{P_{(t)}}{P_s} \text{ or } \left[P_{(t)}\right] = 1 - \frac{P_{(t)}}{P_s} \quad (9)$$

$$DFS = t_y, \text{ if } S_{sen}(t) \geq Y_{crit} \quad (10)$$

where $S_{sen}(t)$ represents the state of leaf coloring on day $t$, while $R_{sen}(t)$ denotes its rate. $T_{(t)}$ is the average temperature for day $t$, with $T_b$ as the base temperature. $P_{(t)}$ refers to the $P_{(t)}$ on day $t$, and $P_s$ is the critical threshold for photoperiod. The parameters $x$ and $y$ range between 0 and 2. Leaf coloration was initiated on $DOY_s$ when the daily temperature dropped to or below $T_b$ and the photoperiod shortened to $P_s$. The process continued until day $t_y$, when $S_{sen}(t)$ accumulated to a predefined threshold ($Y_{crit}$). The day $t_y$ was then identified as the DFS.

SIAM and DMT are modified from DM, with $Y_{crit}$ linearly related to the LUD anomaly in SIAM, and to spring–summer temperature in DMT:

$$Y_{crit} = a + b \times LUD_a \quad (11)$$

$$Y_{crit} = a + b \times Tss \quad (12)$$

where $LUD_a$ is the LUD anomaly and Tss is the mean spring-summer temperature.

We constructed a model coefficient, $ALANin_i$, which included the annual ALAN values, and incorporated $ALANin_i$ into the calculation of the forcing rate in the DFS model:

$$ALANin_i = e^{k \times \left(\frac{ALAN_i - ALAN_{max}}{ALAN_{max}}\right)} \quad (13)$$

where $ALAN_i$ represents the ALAN value in year $i$ and $ALAN_{max}$ represents the maximum ALAN value.

To improve the predictive accuracy of the model, we employed the Particle Swarm Optimization algorithm to optimize key model parameters (such as $T_{th}$, $T_b$, $P_s$, $Y_{crit}$, $a$, $b$). This algorithm simulates the movement and collaborative search of particles within the parameter space to find the optimal combination of parameters that minimizes model error in a multi-dimensional space. This algorithm is known for its fast convergence and strong global search capability, making it particularly suitable for nonlinear and multi-parameter coupled problems[71]. The resulting optimal parameter set was then used for subsequent model training and validation.

The metrics for evaluating the accuracy between the predicted and observed DFS included the $R$, the proportion of observations with significant at $P < 0.05$, RMSE and AIC. AIC is used to balance the model's goodness of fit and complexity. A smaller AIC value indicates that the model achieves a good fit while maintaining lower complexity, making it more concise and efficient[72].

$$AIC = n \times \ln\left(\frac{\sum_{i=1}^{n} (p_i - o_i)^2}{n}\right) + 2k \quad (14)$$

where $n$ is the total number of years, $k$ is the number of parameters in the model, $p_i$ and $o_i$ represent the predicted and observed DFS in year $i$, respectively.

To obtain future ALAN data, we performed ordinary least squares regression (Eq. (15)) on the observed ALAN values using each city's average ALAN from 2001 to 2022 and annual GDP data. Model accuracy was evaluated using five-fold cross-validation (Supplementary Fig. 18), and the resulting estimates were used for subsequent analyses.

$$ALAN_i = a \times ALAN_{mean} + b \times \ln(GDP_i) + c \quad (15)$$

where $ALAN_{mean}$ is the average ALAN, and $GDP_i$ is the GDP in year $i$, $a$, $b$, and $c$ are the coefficients of the regression equation.

### Reporting summary

Further information on research design is available in the Nature Portfolio Reporting Summary linked to this article.

## Data availability

All data used in this study are freely available from the following sources: In situ DFS data can be accessed from https://doi.org/10.5281/zenodo.17925641 and http://www.pep725.eu/. Satellite-derived DFS data is available from https://lpdaac.usgs.gov/products/mcd12q2v061/. NPP-VIIRS-like nighttime light data is available from https://dataverse.harvard.edu/dataset.xhtml?persistentId=doi:10.7910/DVN/YGIVCD. DMSP-OLS nighttime light data is available from https://eogdata.mines.edu/products/dmsp/. NPP-VIIRS nighttime light data is available from https://eogdata.mines.edu/products/vnl/. H-NTL-v2 data is available from https://doi.org/10.5281/zenodo.17925641. Global urban boundaries data is available from https://data-starcloud.pcl.ac.cn/iearthdata/map?id=14. Six-hourly temperature data is available from https://catalogue.ceda.ac.uk/uuid/aed8e269513f446fb1b5d2512bb387ad/. Monthly climatic data is available from https://www.climatologylab.org/terraclimate.html. HDI, GDP, Per capita GDP data is available from https://datadryad.org/dataset/doi:10.5061/dryad.dk1j0. Aboveground biomass is available from https://zenodo.org/records/13331493. Tree density is available from https://elischolar.library.yale.edu/yale_fes_data/

1/. Canopy height is available from https://webmap.ornl.gov/ogc/dataset.jsp?ds_id=1665. GPP, ET, FPAR data are available from https://lpdaac.usgs.gov/products/. $V_{cmax}$ data is available from https://www.nesdc.org.cn/sdo/detail?id=612f42ee7e28172cbed3d80f. SIF is available from https://globalecology.unh.edu/data/GOSIF.html. Future temperatures, Per capita GPP data were from the CMIP6 models (https://esgf-node.llnl.gov/projects/esgf-llnl/). Source data are provided with this paper.

## Code availability

All data analyses and modeling were performed using Python (v3.8.10). The code is stored in a publicly available Zenodo repository https://doi.org/10.5281/zenodo.17925641.

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

## Acknowledgements

This study was supported by National Natural Science Foundation of China grant (42125101, W2412014). W.Q. was supported by National Natural Science Foundation of China grant (42401029). J.P. was supported by the Spanish Government grants TED2021-132627 B–I00 and PID2022-140808NB-I00, funded by MCIN, AEI/10.13039/ 501100011033 European Union Next Generation EU/PRTR, and the Fundación Ramón Areces grant CIVP20A6621.

## Author contributions

C.W. designed the research. Y.C. and W.Q. wrote the first draft of the manuscript. Y.C. performed the analyses and visualization. W.Q. processed ground-based phenology datasets. C.M.Z. and J.P. discussed the design, methods and results and substantially revised the manuscript with intensive suggestions.

## Competing interests

The authors declare no competing interests.
