## [Peer Review file · Nature Communications]

Increased artificial illumination delays urban autumnal foliar senescence

Corresponding Author: Professor Chaoyang Wu

Version 0:

Reviewer comments:

Reviewer #1

(Remarks to the Author)

This manuscript studies how artificial light at night (ALAN) delays Dates of Foliar Senescence (DFS) using nighttime light remote sensing data and phenology metrics from field and satellite observations. The manuscript is overall well written.

I have two critical concerns:

1. About the nighttime light data. This study is built upon the "NPP-VIIRS-like" nighttime light product. This product is generated by "calibrating" the DMSP-OLS nighttime light data - which has no radiometric calibration - to VIIRS nighttime light data.

I think it is problematic. DMSP-OLS data is inherently not radiometrically calibrated, which means that the nighttime light intensity (ALAN intensity) detected by DMSP-OLS is unit-less and it not comparable over time. The calibrated "NPP-VIIRS-like" nighttime light data is created based on a few assumptions on how DMSP-OLS based nighttime intensity changes over time, as well as on the relationship between DMSP-OLS and VIIRS nighttime light data.

That is to say: (1) the resulting nighttime light intensity of "NPP-VIIRS-like" nighttime light product is sensitive to those above-mentioned assumptions during the calibration process. and (2) most importantly, the resulting "NPP-VIIRS-like" nighttime light intensity generated from DMSP-OLS is still unit-less and not suitable for quantitative analysis involving nighttime light radiance values (how can we give assign radiance values to a data without radiometric calibration?)

Thus, the results derived from this questionable data are not robust and convincing.

2. About the findings on Fig.2. It seems that the finding indicates that "Development" based variables, such as human development index and ALAN, are dominant contributors of DFS. This is quite different from existing literatures, which suggests temperature is the key factors of DFS. Why?

Reviewer #2

(Remarks to the Author)

This manuscript aims to quantify the effects of artificial light at night (ALAN) on date of foliar senescence (DFS) in Europe and China through a combination of satellite observation of ALAN and in-situ observations of foliar senescence.

Results indicated that the effect of ALAN on DFS was more pronounced for areas with drier and warmer climates and that human development variables were disproportionately influential on DFS. The manuscript does a good job at testing the direct causal relationships of ALAN and multiple other climate variables on DFS to determine the relative importance of

different variables and identify that ALAN has a disproportionate effect on DFS.

Findings are all supported by the data and analyses. The significant improvements in DFS models based on in-situ and satellite data represent a substantial improvement in our ability to understand and model/predict environmental effects of climate change and increasing human development. Overall, this manuscript is well written and will make a significant contribution to environmental science and policy.

Reviewer #3

(Remarks to the Author)

This manuscript addresses an important and timely topic: the influence of artificial light at night (ALAN) on the timing of autumn phenology (date of foliar senescence, DFS). The dataset appears strong and the research question highly relevant. However, in its current form, the potential insights are buried under a complex mix of statistical methods that are not presented in an accessible way, which makes it difficult to assess the value of the research.

A substantial part of the thematic ground has already been covered by Wang et al. (Nature Cities, 2025 and see also Du et al. 2022, International Journal of Applied Earth Observation and Geoinformation), which is only marginally cited here. That prior study not only overlaps with the present work, but also presents the findings much clearer. The novelty and added value of the current manuscript are therefore insufficiently established.

Moreover, the attempt to test physiological mechanisms (e.g. via PCMCi+) is unconvincing. These methods can suggest statistical associations but cannot provide mechanistic evidence at the global scale. In addition, the hypotheses presented in the manuscript appear somewhat ad hoc and not well anchored either in the empirical results (e.g. HDI emerging as the strongest predictor) or in established theory. In my view, this whole section should be removed.

To me, the most promising direction lies in the extension of process-based autumn phenology models (CDD, DM, SIAM, DMT) with ALAN effects. This is a novel approach and could be a valuable contribution if developed rigorously. However, that would require a clearer focus and would essentially constitute a different manuscript.

K. Schiffers

Reviewer #4

(Remarks to the Author)

Version 1:

Reviewer comments:

Reviewer #1

(Remarks to the Author)

The authors have well addressed my concerns.

(Remarks on code availability)

Reviewer #3

(Remarks to the Author)

We have now read through the revised version of the manuscript. The methodological framework and the underlying hypotheses become somewhat clearer once the reader has worked through the expanded Supplementary Materials, and the additional clarification regarding the HDI-related interpretation is helpful and improves the coherence of the results. These revisions certainly strengthen the manuscript.

That said, the description of the methodological approach in the main text remains complex. Although we are familiar with tree phenology data and modeling, the methodological pipeline is so intricate that we find it difficult to fully judge whether all methodological choices are appropriate or optimally implemented. While the authors have clarified some of the limitations, the overall structure of the analysis remains dense and challenging to follow for readers without deep expertise in these specific techniques. The DFS models are implemented as a single jupyter notebook with more than 4000 lines, which makes it extremely difficult to review and to understand the analytical workflow. It would greatly improve transparency and usability if the code were modularized or separated into smaller components.

Also, we are still not entirely comfortable with the way the term causal is used. The authors state that they “verify the causal influence of ALAN on the DFS” (line 399). Given that PCMCi+ identifies statistical relationships (which becomes clearer now in other parts of the manuscript), I would encourage the authors to use more precise wording, e.g., referring to statistical or Granger-causal influence, so as not to overstate what the method can demonstrate.

(Remarks on code availability)

We did not check the content of the whole code but noticed that several aspects could be improved to enhance transparency and reproducibility. In particular, the notebooks contain hard-coded absolute Windows paths (e.g., H:\...), which will not work for users on other systems or when accessing the code from GitHub. I recommend replacing these with relative paths within the repository structure. In addition, some imports are redundant or unused, and the structure of the code would benefit from clearer comments and more explicit use of column names rather than positional indexing.

Reviewer #4

(Remarks to the Author)

(Remarks on code availability)

Reviewer #1 (Remarks to the Author):

This manuscript studies how artificial light at night (ALAN) delays Dates of Foliar Senescence (DFS) using nighttime light remote sensing data and phenology metrics from field and satellite observations. The manuscript is overall well written.

Response: We sincerely appreciate your positive feedback. Your comments and suggestions are important to us, and we have carefully addressed them in our revisions, as detailed below.

I have two critical concerns:

1. About the nighttime light data. This study is built upon the "NPP-VIIRS-like" nighttime light product. This product is generated by "calibrating" the DMSP-OLS nighttime light data - which has no radiometric calibration - to VIIRS nighttime light data.

I think it is problematic. DMSP-OLS data is inherently not radiometrically calibrated, which means that the nighttime light intensity (ALAN intensity) detected by DMSP-OLS is unit-less and it not comparable over time. The calibrated "NPP-VIIRS-like" nighttime light data is created based on a few assumptions on how DMSP-OLS based nighttime intensity changes over time, as well as on the relationship between DMSP-OLS and VIIRS nighttime light data.

That is to say: (1) the resulting nighttime light intensity of "NPP-VIIRS-like" nighttime light product is sensitive to those above-mentioned assumptions during the calibration process. and (2) most importantly, the resulting "NPP-VIIRS-like" nighttime light intensity generated from DMSP-OLS is still unit-less and not suitable for quantitative analysis involving nighttime light radiance values (how can we give assign radiance values to a data without radiometric calibration?)

Thus, the results derived from this questionable data are not robust and convincing.

Response: We appreciate the reviewer's valuable comment regarding data quality and reliability. The ALAN dataset used in our analysis (2001-2022) is the NPP-VIIRS-like nighttime light (NTL) product, which was generated through a rigorous cross-sensor calibration between DMSP-OLS and NPP-VIIRS observations (Chen et al., 2021). The calibrated DMSP-OLS data for 2001-2013 were derived from the original annual DMSP-

OLS NTL data using a stepwise calibration approach (Li and Zhou, 2017), ensuring high temporal consistency and strong agreement with NPP-VIIRS data for 2014-2022 (Li et al., 2020). Validation analyses demonstrated high reliability of this harmonized dataset, with strong correlations at both the pixel ($R^2 = 0.87$, $RMSE = 2.96 \text{ nW cm}^{-2} \text{ sr}^{-1}$) and city ($R^2 = 0.94$) levels (Chen et al., 2021). The dataset has been widely adopted across disciplines, including studies on global forest fragmentation driven by urbanization (Ma et al., 2023), ALAN effects on bird diversity (Sun et al., 2022), global nitrogen deposition drivers (Zhu et al., 2025), hydrology-induced migration patterns (Qiao et al., 2024), and global urban heat island intensity (Yang et al., 2024). These extensive applications further demonstrate the robustness and versatility of the NPP-VIIRS-like NTL data.

To further confirm the robustness of our findings, we compared the DFS responses to ALAN derived from four independent NTL datasets: (1) the NPP-VIIRS-like product (used in this study, 2001-2022), (2) the original DMSP-OLS data (2001-2013), (3) the original NPP-VIIRS data (2014-2022), and (4) a recently released long-term DMSP-OLS-like NTL dataset calibrated from NPP-VIIRS observations (2001-2022, Geng et al., 2025). The results were highly consistent across all datasets (Figures 1 and 2 below). Each dataset showed that ALAN generally delays DFS, exhibiting similar spatial distribution patterns. Moreover, all datasets revealed a nonlinear relationship between SV_{ALAN} and ALAN intensity: at low ALAN levels, the delaying effect on DFS was pronounced but weakened and saturated under high-intensity lighting (Figure 1). Using the DMSP-OLS-like dataset (Geng et al., 2025), we further confirmed that 67.5% of cities experienced delayed DFS, and the spatial pattern of responses closely matched that from the NPP-VIIRS-like product (Figure 2). A direct comparison of SV_{ALAN} derived from the two harmonized datasets also showed strong agreement ($R^2 = 0.67$, $RMSE = 0.09$), with 82.5% of cities exhibiting consistent sensitivity directions.

In summary, the NPP-VIIRS-like product (Chen et al., 2021) is based on robust cross-sensor calibration and extensive validation, with proven reliability through widespread scientific applications. Our sensitivity analyses using multiple independent or calibrated NTL datasets confirm that the key findings of our study are highly robust to data source

and calibration choices. We have clarified these methodological details and supporting results in the revised manuscript (Lines 104-107, 312-320, Supplementary Figs. 6 and 7).

Figure 1. Spatial patterns and intensity responses of the sensitivity (SV_{ALAN}) of the date of foliar senescence (DFS) to the intensity of artificial light at night (ALAN). Spatial distribution of SV_{ALAN} (a-c) and differences in SV_{ALAN} between low- and high-ALAN intensity regions (d-f) derived from the original and two comparative analyses.

Figure 2. Comparison of the sensitivity (SV_{ALAN}) of the date of foliar senescence

(DFS) to the intensity of artificial light at night (ALAN) derived from two nighttime light datasets. Spatial patterns of SV_{ALAN} estimated from the NPP-VIIRS-like (a) and H-NTL-v2 datasets (b), respectively. c, Differences in SV_{ALAN} between low- and high-ALAN intensity regions based on the H-NTL-v2 dataset. d, Linear relationship of SV_{ALAN} values derived from the two datasets; the red dashed line indicates the 1:1 reference line. e, Frequency of matched SV_{ALAN} sign patterns between the two datasets, where “++” and “--” indicate consistent positive and negative sensitivities, and “+-” and “-+” indicate opposite sensitivities.

References:

- Chen, Z. et al. An extended time series (2000–2018) of global NPP-VIIRS-like nighttime light data from a cross-sensor calibration. *Earth Syst. Sci. Data* **13**, 889–906 (2021).
- Li, X., Zhou, Y., Zhao, M., & Zhao, X. A harmonized global nighttime light dataset 1992 – 2018. *Sci. Data* **7**, 168 (2020).
- Geng, M. et al. An efficient method for aurora and noise reduction with a harmonized nighttime light dataset. *Remote Sens. Environ.* **328**, 114891 (2025).
- Ma, J., Li, J., Wu, W., & Liu, J. Global forest fragmentation change from 2000 to 2020. *Nat. Commun.* **14**, 3752 (2023).
- Qiao, R. et al. Understanding the global subnational migration patterns driven by hydrological intrusion exposure. *Nat. Commun.* **15**, 6285 (2024).
- Sun, B. et al. Urbanization affects spatial variation and species similarity of bird diversity distribution. *Sci. Adv.* **8**, eade3061 (2022).
- Yang, Q. et al. A global urban heat island intensity dataset: Generation, comparison, and analysis. *Remote Sens. Environ.* **312**, 114343 (2024).
- Zhu, J. et al. Changing patterns of global nitrogen deposition driven by socio-economic development. *Nat. Commun.* **16**, 46 (2025).

2. About the findings on Fig.2. It seems that the finding indicates that "Development" based variables, such as human development index and ALAN, are dominant contributors of DFS. This is quite different from existing literatures, which suggests temperature is the key factors of DFS. Why?

Response: We apologize for the misunderstanding in the original version. The spatial analysis presented in Fig. 2 focuses on identifying the factors driving the spatial variability in ALAN effects on DFS (i.e., the sensitivity of DFS to ALAN), rather than the factors directly controlling DFS itself (see Figure 3 below). We fully agree with the reviewer that temperature is the dominant factor regulating the overall variation in DFS. Our mechanistic analysis further suggests that ALAN may modulate the phenological response of vegetation to temperature, indicating an ALAN's indirect influence rather than a direct control on DFS. We have revised the relevant text in the manuscript to clarify this distinction and prevent potential confusion (Lines 121-123, 134-135).

Figure 3. Spatial attribution analysis of artificial light at night (ALAN) effects on the date of foliar senescence (DFS). **a**, The relative importance of factors controlling the spatial variability of ALAN effects on DFS (i.e., SV_{ALAN}), determined by a random-forest model ($R^2 = 0.83$, $n = 438$) using mean absolute SHAP values. The inner subplot indicates the average importance of factors associated with development, vegetation, and climate. **b**, The right beeswarm plot shows the distribution of SHAP values for each factor.

Thank you again for your detailed suggestions and comments, which highly improved our manuscript.

Reviewer #2 (Remarks to the Author):

This manuscript aims to quantify the effects of artificial light at night (ALAN) on date of foliar senescence (DFS) in Europe and China through a combination of satellite observation of ALAN and in-situ observations of foliar senescence.

Results indicated that the effect of ALAN on DFS was more pronounced for areas with drier and warmer climates and that human development variables were disproportionately influential on DFS. The manuscript does a good job at testing the direct causal relationships of ALAN and multiple other climate variables on DFS to determine the relative importance of different variables and identify that ALAN has a disproportionate effect on DFS.

Findings are all supported by the data and analyses. The significant improvements in DFS models based on in-situ and satellite data represent a substantial improvement in our ability to understand and model/predict environmental effects of climate change and increasing human development. Overall, this manuscript is well written and will make a significant contribution to environmental science and policy.

Response: We sincerely thank you for the positive and encouraging comments. We have improved the manuscript based other reviewers to ensure clarity and strengthen its overall quality.

Reviewer #3 (Remarks to the Author):

This manuscript addresses an important and timely topic: the influence of artificial light at night (ALAN) on the timing of autumn phenology (date of foliar senescence, DFS). The dataset appears strong and the research question highly relevant. However, in its current form, the potential insights are buried under a complex mix of statistical methods that are not presented in an accessible way, which makes it difficult to assess the value of the research.

Response: Thank you very much for your detailed comments and suggestions. We have carefully addressed them in our revisions, as detailed below.

A substantial part of the thematic ground has already been covered by Wang et al. (Nature Cities, 2025 and see also Du et al. 2022, International Journal of Applied Earth Observation and Geoinformation), which is only marginally cited here. That prior study not only overlaps with the present work, but also presents the findings much clearer. The novelty and added value of the current manuscript are therefore insufficiently established.

Response: We agree that an increasing number of studies have examined the influence of ALAN on plant phenology, such as Wang et al. (2025) and Du et al. (2022) you mentioned here. We acknowledge that these studies have provided valuable insights into the effects of ALAN on DFS, advancing our understanding of carbon cycling and climate adaptation mechanisms in urban ecosystems. However, our study differs from these works in both research design, findings, and implications, which we have now clarified in the revised manuscript.

Research design and framework: Du et al. (2022) used ALAN intensity as a proxy for urbanization (without investigating the direct influence of ALAN) to assess how climate change and urbanization jointly affect DFS, while Wang et al. (2025) primarily compared the relative contributions of ALAN and the urban heat island (UHI) effect to delayed DFS (lack of mechanistic exploration). Unlike these studies, which focus mainly on empirical relationships between ALAN/urbanization and DFS, our study adopts a **phenomenon-mechanism-model** framework that links observed phenological shifts to underlying

physiological and ecological processes under ALAN exposure and integrates ALAN effects into predictive DFS models. A key contribution of our work is bridging the gap between observation, mechanism, and modeling, thereby providing mechanistic insights not captured in earlier studies.

Research findings and implications: Our results confirmed the overall delaying effect of ALAN on DFS, consistent with Wang et al. (2025). Beyond this, we identified a **nonlinear ALAN effect**, whereby low-intensity light delays DFS, while high-intensity light diminishes or reverses this delay. This finding is particularly important given the accelerating global urbanization and climate change. Furthermore, we identified two potential mechanistic pathways, i.e., enhanced carbon assimilation and altered climatic sensitivities, and demonstrated that incorporating ALAN effects into phenology models substantially improves predictive performance. Together, these findings extend previous work by offering mechanistic explanations and model-based predictions across broader spatial and temporal scales.

We have carefully revised the manuscript to highlight these unique contributions as suggested (Lines 61-73, 217-218).

References:

Du H. et al. Responses of autumn vegetation phenology to climate change and urbanization at northern middle and high latitudes. *Int. J. Appl. Earth Obs. Geoinf.* **115**, 103086 (2022).

Wang, L. et al. Artificial light at night outweighs temperature in lengthening urban growing seasons. *Nat Cities* **2**, 506-517 (2025).

Moreover, the attempt to test physiological mechanisms (e.g. via PCMCi+) is unconvincing. These methods can suggest statistical associations but cannot provide mechanistic evidence at the global scale.

Response: Thank you for raising this important point. We agree that PCMCi+ itself does not provide direct physiological evidence; rather, it serves as a powerful statistical framework for inferring potential causal relationships among interdependent variables. The strength of PCMCi+ lies in its ability to account for confounding influences from other variables, thereby isolating specific causal links more reliably. This method has been

widely adopted across disciplines to uncover causal structures in high-dimensional time-series data and has proven effective in identifying both lagged and contemporaneous dependencies. For instance, PCMCI+ has been used to detect causal links between nitrogen deposition and DFS (Wang et al., 2025), to disentangle coupled atmospheric-oceanic-biogeochemical interactions in the subpolar North Atlantic (Bénard et al., 2024), and to identify region-invariant predictors of supraglacial lake evolution in Greenland via the RIC-TSC framework, demonstrating its superiority over conventional correlation-based methods under nonstationary conditions (Hossain et al., 2025).

In our study, we first explored the associations among ALAN, photosynthetic proxies, and DFS using partial correlation analysis, and subsequently applied PCMCI+ to test whether potential causal relationships exist among these variables (see Figure below). Therefore, PCMCI+ was not used to directly claim physiological causality, but rather to assess whether the inferred statistical causality aligns with established ecological mechanisms. Thus, we have retained the PCMCI+ approach in the revised manuscript and clarified its role and rationale accordingly (Lines 400-413, Supplementary Fig. 10)

Figure. The framework of casual analysis using the PCMCI+ method.

References:

- Wang, J., Wang, X., Peñuelas, J., Hua, H., & Wu, C. Nitrogen deposition favors later leaf senescence in woody species. *Nat. Commun.* **16**, 3668 (2025).
- Bénard, G., Gehlen, M., & Vrac, M. A causality-based method for multi-model comparison: Application to relationships between atmospheric and marine biogeochemical variables. *Earth Syst. Dynam. Discuss.* **2024**, 1-26 (2024).
- Hossain, E., Ferdous, M. H., Dunmire, D., Subramanian, A., & Gani, M. O. Causal Time Series Modeling of Supraglacial Lake Evolution in Greenland under Distribution Shift. *arXiv preprint arXiv:2510.15265* (2025).

In addition, the hypotheses presented in the manuscript appear somewhat ad hoc and not well anchored either in the empirical results (e.g. HDI emerging as the strongest predictor)

or in established theory. In my view, this whole section should be removed.

Response: We appreciate the reviewer's comment and acknowledge that our original description may have been unclear. As noted above, mechanistic exploration is one of the key contributions of our study. The hypotheses we proposed were developed through an extensive review and synthesis of existing literature (see Table below), encompassing both manipulative experiments (site-scale) and satellite-based observations (region-scale). This literature indicates that ALAN can affect plant carbon assimilation through photosynthetic processes and alter the climatic sensitivity of DFS. Experimental studies across multiple species have demonstrated that ALAN can sustain or modify nocturnal photosynthetic activity though its effects vary by species, potentially enhancing photosynthesis. ALAN can also alter soluble sugar and chlorophyll content, thereby influencing net photosynthetic production. In addition, ALAN modulates plant responses to climatic variables such as temperature and precipitation. By artificially extending perceived day length, ALAN may disrupt plants' adaptive responses to temperature and precipitation variability, supporting our hypothesis that ALAN interacts with climatic drivers to co-regulate phenological transitions.

Based on these empirical evidences, we explicitly tested these two mechanistic pathways by integrating multiple datasets and applying complementary statistical approaches. Overall, our results support both hypothesized pathways and provide mechanistic insights into how enhanced ALAN exposure delays DFS. To improve clarity, we have included a literature summary in the Supplementary Materials to support the formulation of our hypotheses and updated the relevant figure (see below) to illustrate our mechanistic framework more clearly. Regarding the HDI, while it emerged as the strongest predictor in the spatial attribution analysis, this result does not directly explain temporal responses of DFS to ALAN increases within the same region. In the revised manuscript, we have acknowledged the limitations of mechanistic exploration and emphasized the importance of incorporating field experiments to measure hormonal changes (e.g., auxin, ethylene, and abscisic acid) in future work. Corresponding revisions and supporting

references have been added (Lines 142-170, 172-179, 235-237, 245-257, Fig. 3, Supplementary Fig. 13, Supplementary Table 2).

Figure. Mediating pathways linking artificial light at night (ALAN) to the date of autumnal foliar senescence (DFS). **a-b**, Proportions of cities with significantly positive, negative, and nonsignificant partial correlations between ALAN and photosynthesis (outer rings) and between photosynthesis and DFS (inner rings) ($P < 0.05$, two-tailed t-test). Photosynthetic indicators include the maximum rate of carboxylation (V_{cmax}) (**a**), solar-induced chlorophyll fluorescence (SIF) (**b**). **c**, The modulating effect of ALAN on climatically driven DFS responses. **d**, Conceptual diagram illustrating potential mechanisms by which ALAN influences DFS variations.

Table. Literature support for the proposed mechanistic pathways

	Reference	Light type	Species type	Method	Conclusion	DOI
H1: Carbon sequestration	Kim et al. 2015	High-pressure sodium (HPS)	Cymbidium (Red Fire and Yokihi)	Manipulative experiment. Plants in the control group grew under natural daylight conditions, while those in the treatment groups were exposed to artificial light.	Cymbidium (Red Fire and Yokihi) orchids were able to carry out photosynthesis under artificial light at night.	doi.org/10.1016/j.scienta.2015.01.036
	Ermes et al.,2021	Light-emitting diode (LED)	Tilia plathyphyllos Scop. and Platanus x acerifolia (Aiton) Willd	Manipulative experiment. The control group grew under natural dark conditions at night, while the experimental group was exposed to artificial light at night to simulate urban street lighting conditions.	Plants exposed to light exhibited positive nighttime photosynthetic rates, indicating active CO ₂ assimilation compared with the control group. Moreover, chlorophyll content was higher in the plants exposed to light.	doi.org/10.12871/00021857202146
	Czaja et al.,2022	LED	Cornus alba and Lonicera pileata	Manipulative experiment. Three treatments were applied: control (12 h light/12 h dark), moderate light pollution (12 h light/12 h dim light), and low light pollution (12 h light/12 h intermittent light).	Light pollution treatment increased soluble sugar content in the apical twigs of Cornus alba and Lonicera pileata compared with the control.	doi.org/10.1016/j.ufug.2022.127753

	Ermes et al.2024	LED	white poplar trees (Populus alba clone DI-1)	Manipulative experiment. The control group received no illumination after dusk, while the treatment group was exposed to street lighting at night.	Nighttime lighting allowed plants to maintain photosynthesis at night, whereas the control group primarily exhibited respiration at night.	doi.org/10.1016/j.envexpbot.2024.105861
	Kolman et al. 2025	LED	Common alder (Alnus glutinosa (L.) Gaertn.), Elder (Sambucus nigra L.), Field maple (Acer campestre L.), Silver birch (Betula pendula Roth), Tree-of-heaven (Ailanthus altissima (Mill.) Swingle)	Manipulative experiment. Branches under streetlights and leaves from the opposite (control) side of the same trees, which received less artificial light at night, were collected for laboratory analysis.	Under strong light pollution, the fluorescence yield increased significantly in common linden and elder, and field maple, silver birch, and tree-of-heaven showed higher electron transport efficiency under street lighting.	doi.org/10.1002/pei3.70032
H2: Climate responses	Bennie et al.2018	LED	A. tenuis , Anthoxanthum odoratum and H. lanatus	Manipulative experiment. The control group vegetation was not exposed to artificial light, while the experimental groups were treated with broad-spectrum cool white LED light, and with near-monochromatic amber LED light.	When temperature cues become unreliable, plants rely more on photoperiod. Artificial light extends perceived day length, disrupting this balance and altering the strength of phenological responses to temperature variations	doi.org/10.1111/1365-2664.12927

Meng et al. 2022	NPP-VIIRS DNB (500m)	NA	Remote sensing – based study. Using remote-sensing-derived Artificial light at night (ALAN) data and phenological network observations, compared phenological values under conditions with and without ALAN.	ALAN and temperature interactively influence phenology. At lower temperatures, ALAN may amplify the autumn phenological delay caused by warming, whereas at higher temperatures, ALAN may weaken or even reverse this delay.	doi.org/10.1093/pnasnexus/pgac046
Du et al. 2022	Zhao et al. 2020 (1km)	NA	Remote sensing – based study. Using remote-sensing-derived ALAN data, phenology data, and meteorological data, the differences in the effects of temperature and precipitation on autumn phenology were assessed under varying levels of ALAN.	Different levels of urbanization affect the response of the end of the growing season (EOS) to changes in temperature and precipitation.(with urbanization level represented by nighttime light intensity).	doi.org/10.1016/j.jag.2022.103086 doi.org/10.1109/TGRS.2019.2949797
Wang et al. 2025	NPP-VIIRS DNB (500m)	NA	Remote sensing – based study. Using remote-sensing-derived phenology data, ALAN data, and temperature data, partial correlation analysis was conducted to assess the effects of ALAN and temperature on phenology.	ALAN played a more important role in delaying EOS than air temperature, but its effect diminished as city temperatures increased.	doi.org/10.1038/s44284-025-00258-2

To me, the most promising direction lies in the extension of process-based autumn phenology models (CDD, DM, SIAM, DMT) with ALAN effects. This is a novel approach and could be a valuable contribution if developed rigorously. However, that would require a clearer focus and would essentially constitute a different manuscript.

K. Schiffers

Response: Thank you for your recognition of our work on phenological modeling. Our study aims to systematically elucidate the effects of ALAN on autumn phenology, including its observed patterns, underlying mechanisms, and model improvement. The preliminary work has laid an important foundation for the subsequent model enhancement. We agree that incorporating the effects of ALAN into autumn phenology process models could enable more in-depth research. We have improved the description of DFS modelling and updated the schematic diagram of model parameters (see Figure below) (Lines 267-269, Supplementary Fig. 14).

Figure. Schematic diagram of the input parameters for four original DFS models, including the semi-empirical model based on the cooling degree days (CDD), the Delpierre model (DM), the spring-influenced autumn DFS model (SIAM), and the DM modified by spring-summer temperature (DMT), as well as their ALAN-improved versions (CDD_{ALAN}, DM_{ALAN}, SIAM_{ALAN}, DMT_{ALAN}). LUD denotes the leaf-unfolding date, and TSS represents the mean spring-summer temperature.

Thank you again for your detailed suggestions and comments, which highly improved our manuscript.

Reviewer #4 (Remarks to the Author):

Response: Thank you very much for your contribution to our work.

We express our gratitude to the reviewers for their valuable comments and suggestions, which have significantly enhanced the coherence and rigor of our study. We have thoroughly addressed each comment and suggestion in the revised manuscript. Please find our detailed responses to each point below.

Reviewer #1 (Remarks to the Author):

The authors have well addressed my concerns.

Response: We sincerely appreciate your contribution in improving our study. Thanks again for your positive feedback and recognition of our study's significance.

Reviewer #3 (Remarks to the Author):

We have now read through the revised version of the manuscript. The methodological framework and the underlying hypotheses become somewhat clearer once the reader has worked through the expanded Supplementary Materials, and the additional clarification regarding the HDI-related interpretation is helpful and improves the coherence of the results. These revisions certainly strengthen the manuscript.

Response: We appreciate your positive feedback on our revisions, and your further comments in improving our study.

That said, the description of the methodological approach in the main text remains complex. Although we are familiar with tree phenology data and modeling, the methodological pipeline is so intricate that we find it difficult to fully judge whether all methodological choices are appropriate or optimally implemented. While the authors have clarified some of the limitations, the overall structure of the analysis remains dense and challenging to follow for readers without deep expertise in these specific techniques. The DFS models are implemented as a single jupyter notebook with more than 4000 lines, which makes it extremely difficult to review and to understand the analytical workflow. It would greatly improve transparency and usability if the code were modularized or separated into smaller components.

Response: Thank you for your feedback. We have modularized the original 4000+ line Jupyter notebook by separating it into smaller, functionally distinct components, thereby

improving code clarity and usability.

Also, we are still not entirely comfortable with the way the term causal is used. The authors state that they “verify the causal influence of ALAN on the DFS” (line 399). Given that PCMCI+ identifies statistical relationships (which becomes clearer now in other parts of the manuscript), I would encourage the authors to use more precise wording, e.g., referring to statistical or Granger-causal influence, so as not to overstate what the method can demonstrate-e.

Response: We thank you for this important point regarding the precise use of causal terminology. We agree that our previous wording could be interpreted as overstating what PCMCI+ can demonstrate. In the revised manuscript, we have carefully adjusted our language to more accurately reflect the nature of the relationships identified by PCMCI+. Lines 367–384.

Reviewer #3 (Remarks on code availability):

We did not check the content of the whole code but noticed that several aspects could be improved to enhance transparency and reproducibility. In particular, the notebooks contain hard-coded absolute Windows paths (e.g., H:\...), which will not work for users on other systems or when accessing the code from GitHub. I recommend replacing these with relative paths within the repository structure. In addition, some imports are redundant or unused, and the structure of the code would benefit from clearer comments and more explicit use of column names rather than positional indexing.

Response: We thank you for these helpful suggestions on improving code transparency and reproducibility. We have implemented all the recommended changes in the updated code repository.

Reviewer #4 (Remarks to the Author):

Response: Thank you very much for your contribution to our work.

ROUND 1 REVIEWER 2 ATTACHMENT:

Increased artificial illumination delays urban autumnal foliar senescence

This manuscript aims to quantify the effects of artificial light at night (ALAN) on date of foliar senescence (DFS) in Europe and China through a combination of satellite observation of ALAN and in-situ observations of foliar senescence.

Results indicated that the effect of ALAN on DFS was more pronounced for areas with drier and warmer climates and that human development variables were disproportionately influential on DFS. The manuscript does a good job at testing the direct causal relationships of ALAN and multiple other climate variables on DFS to determine the relative importance of different variables and identify that ALAN has a disproportionate effect on DFS.

Findings are all supported by the data and analyses. The significant improvements in DFS models based on in-situ and satellite data represent a substantial improvement in our ability to understand and model/predict environmental effects of climate change and increasing human development. Overall, this manuscript is well written and will make a significant contribution to environmental science and policy.

Detailed Feedback:

L54-55: Extra “as an” in this sentence.

L119-130: If the model explains ~83% of the observed spatial variability, it would be good to see some proposed explanation for the remaining 17%. Given that ALAN is such a ubiquitous issue, this would place these results into important context.

L269: “...accuracy across scales (Fig. 4). underscoring the importance...”Period should be comma.

L431: The approach taken here is good, especially because it allows for greater interpretability of the model results, but I wonder about potential non-linear effects/relationships. The analyses used here were based on linear relationships between different variables, but environmental and climatic relationships are not infrequently non-linear. If the authors could address this challenge briefly it would only strengthen the manuscript.